# Topic-DPR: Topic-based Prompts for Dense Passage Retrieval

**Qingfa Xiao[1], Shuangyin Li[1, *], Lei Chen[2, 3]**

South China Normal University

The Hong Kong University of Science and Technology

The Hong Kong University of Science and Technology (Guangzhou)

qingfaxiao@m.scnu.edu.cn, shuangyinli@scnu.edu.cn, leichen@cse.ust.hk

## Abstract

Prompt-based learning's efficacy across numerous natural language processing tasks has led to its integration into dense passage retrieval. Prior research has mainly focused on enhancing the semantic understanding of pre-trained language models by optimizing a single vector as a continuous prompt. This approach, however, leads to a semantic space collapse; identical semantic information seeps into all representations, causing their distributions to converge in a restricted region. This hinders differentiation between relevant and irrelevant passages during dense retrieval. To tackle this issue, we present Topic-DPR, a dense passage retrieval model that uses topic-based prompts. Unlike the single prompt method, multiple topic-based prompts are established over a probabilistic simplex and optimized simultaneously through contrastive learning. This encourages representations to align with their topic distributions, improving space uniformity. Furthermore, we introduce a novel positive and negative sampling strategy, leveraging semi-structured data to boost dense retrieval efficiency. Experimental results from two datasets affirm that our method surpasses previous state-of-the-art retrieval techniques.

## 1 Introduction

Dense Passage Retrieval (DPR), due to its efficacy and efficiency, has gained significant attention recently (Karpukhin et al., 2020; Lee et al., 2020). DPR encapsulates semantic information of queries and passages within a low-dimensional embedding space and measures relevance using cosine distance.

Prompt-based learning is an effective emerging technique for multiple natural language processing tasks (Liu et al., 2021a; Lester et al., 2021; Wang et al., 2022a). This technique uses a task-specific

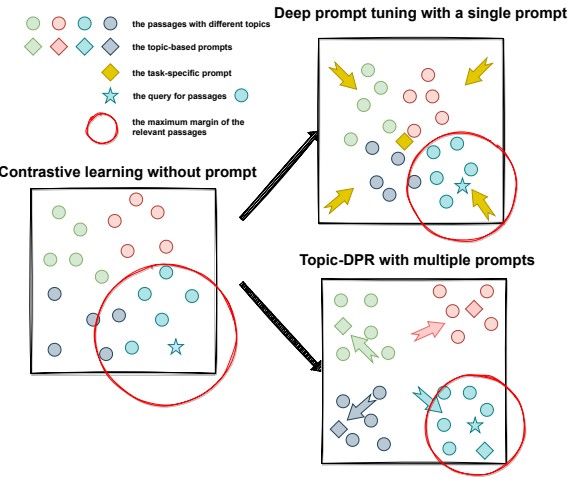

Figure 1: Anisotropic issue of deep prompt tuning with a single prompt. Topical information in prompts aids in distinguishing irrelevant passages in our Topic-DPR.

prompt as input to augment the performance of Pre-trained Language Models (PLMs). Prompts, typically discrete text templates with task-specific information, need explicit definitions. To circumvent local optima during tuning, researchers (Li and Liang, 2021; Liu et al., 2021b) suggested deep prompt tuning that trains a single vector as a continuous prompt. This approach demonstrated effectiveness and flexibility in text-generation tasks. Inspired by deep prompt tuning, recent research (Tang et al., 2022; Tam et al., 2022) has integrated continuous prompts into retrieval tasks. By adding task-specific semantic information as input, these prompts improve PLMs' knowledge utilization and guide PLMs to produce more relevant text representations. Consequently, relevant passage representations are likely closer to the query, thus securing higher rankings.

However, past research has not fully addressed the limitations of a single prompt when dealing with the diverse semantics of a comprehensive dataset. The engagement of all passages with a singular prompt induces a uniform semantic shift.

---

*Corresponding author. Both authors Qingfa Xiao and Shuangyin Li contributed equally to this research.

Imagine using a single prompt like "Explain the main concept of [QUERY]," where [QUERY] is the actual query. While this prompt may effectively steer the Pre-trained Language Model (PLM) to generate representations capturing aspects of each discipline, it may overlook the diverse semantics and terminologies across fields. Consequently, it could have difficulty differentiating articles from distinct disciplines. A single prompt might fall short of capturing the nuances within each discipline, like subtopics or specialized areas. Employing one prompt for all disciplines may result in a uniform semantic shift and a convergence of passage representations in a restricted region of the embedding space, as depicted in Figure 1. This semantic space collapse (Li et al., 2020; Gao et al., 2019; Xiao et al., 2023) can blur the distinction between relevant and irrelevant passages, potentially masking irrelevant passages amidst relevant ones. Therefore, prompt generation is pivotal for this semantically nuanced task. Further analysis is conducted in Section 5.

In this paper, we explore the use of multiple continuous prompts to address the anisotropic issue of deep prompt tuning in dense passage retrieval, a persisting challenge. **Challenge 1: The effective generation of multiple continuous prompts based on the corpus.** A simple approach is partitioning the dataset into subsets via topic modeling, each sharing a common topic-based prompt. This strategy allows the distribution of latent topics across a probabilistic simplex and unsupervised extraction of semantics (Blei et al., 2003; Li et al., 2022b), enabling the definition and initialization of distinct, interpretable topic-based prompts. **Challenge 2: The integration of topic-based prompts into the Pre-trained Language Models (PLMs).** Although our topic-based prompts are defined on a probabilistic simplex using topic modeling, ensuring topical independence, constructing such a simplex and learning topical knowledge within the PLMs' embedding space presents a challenge due to inherent model differences. As a result, we make the topic-based prompts trainable and adopt contrastive learning (Chen et al., 2020; Gao et al., 2021) for optimizing topical relationships.

To tackle these challenges, we introduce a novel framework, Topic-DPR, that efficiently incorporates topic-based prompts into dense passage retrieval. Instead of artificially inflating the number of prompts, we aim to define a prompt set

reflecting the dataset's diverse semantics through a data-driven approach. We propose a unique prompt generation method that utilizes topic modeling to establish the number and initial values of the prompts, which we term topic-based prompts. These prompts are defined within a probabilistic simplex space (Patterson and Teh, 2013), initialized using a topic model such as hierarchical Latent Dirichlet Allocation (hLDA) (Griffiths et al., 2003). Moreover, we propose a loss function based on contrastive learning to preserve the topic-topic relationships of these prompts and align their topic distributions within the simplex. The impact of topic-based prompts serves as a pre-guidance for the PLMs, directing representations towards diverse sub-topic spaces. For dense retrieval, we consider query similarities and design a tailored loss function to capture query-query relationships. We use contrastive learning to maintain query-passage relationships, maximize the similarity between queries and relevant passages, and minimize the similarity between irrelevant pairs. Considering the semi-structured nature of the datasets, we also introduce an in-batch sampling strategy based on multi-category information, providing high-quality positive and negative samples for each query during fine-tuning.

The efficacy of our methods is confirmed through comprehensive experiments, emphasizing the role of topic-based prompts within the Topic-DPR framework. The key contributions are:

1. We propose an unsupervised method for continuous prompt generation using topic modeling, integrating trainable parameters for PLMs adaptation.

2. We introduce Topic-Topic Relation, a novel prompt optimization goal. It uses contrastive learning to maintain topical relationships, addressing the anisotropic issue in traditional deep prompt tuning.

3. Our framework supports the simultaneous use and fine-tuning of multiple prompts in PLMs, improving passage ranking by producing diverse semantic text representations.

## 2 Related Work

### 2.1 Dense Passage Retrieval

Recent advancements in PLMs such as BERT (Devlin et al., 2018), Roberta (Liu et al., 2019), and GPT (Brown et al., 2020) have enabled numerous

unsupervised techniques to derive dense representations of queries and passages for retrieval. These approaches primarily use a Bi-Encoder structure to embed text in a low-dimensional space and learn similarity relations via contrastive learning, contrasting traditional sparse retrieval methods like BM25 or DeepCT (Robertson et al., 2009; Dai and Callan, 2019). DPR (Karpukhin et al., 2020) pioneered an unsupervised dense passage retrieval framework, affirming the feasibility of using dense representations for retrieval independently. This efficient and operational approach was further refined by subsequent studies (Xiong et al., 2020; Gao and Callan, 2021; Ren et al., 2021; Wang et al., 2022b) that focused on high-quality negative sample mining, additional passage relation analysis, and extra training. The essence of these methods is to represent texts in a target space where queries are closer to relevant and distant from irrelevant passages.

## 2.2 Prompt-based Learning

As PLMs, such as GPT-3 (Brown et al., 2020), continue to evolve, prompt-based learning (Gu et al., 2021; Lester et al., 2021; Qin and Eisner, 2021; Webson and Pavlick, 2021) has been introduced to enhance semantic representation and preserve pre-training knowledge. Hence, for various downstream tasks, an effective prompt is pivotal. Initially, discrete text templates were manually designed as prompts for specific tasks (Gao et al., 2020; Ponti et al., 2020; Brown et al., 2020), but this could lead to local-optimal issues due to the neural networks' continuous nature. Addressing this, Li and Liang (2021) and Liu et al. (2021b) highlighted the universal effectiveness of well-optimized prompt tuning across various model scales and natural language processing tasks.

Recent studies have adapted deep prompt tuning for downstream task representation learning. PromCSE (Jiang et al., 2022) uses continuous prompts for semantic textual similarity tasks, enhancing universal sentence representations and accommodating domain shifts. Tam et al. (2022) introduced parameter-efficient prompt tuning for text retrieval across in-domain, cross-domain, and cross-topic settings, with P-Tuning v2(Liu et al., 2021b) exhibiting superior performance. DPTDR (Tang et al., 2022) incorporates deep prompt tuning into dense passage retrieval for open-domain datasets, achieving exceptional performance with minimal parameter tuning.

# 3 The proposed Topic-DPR

## 3.1 Problem Setting

Consider a collection of $M$ documents represented as $D = \{(T_1, A_1, C_1), ..., (T_M, A_M, C_M)\}$, where each 3-tuple $(T_i, A_i, C_i)$ denotes a document with a title $T_i$, an abstract $A_i$, and a set of multi-category information $C_i$. The objective of dense passage retrieval is to find relevant passages $A_j$ for a given query $T_i$, where their multi-category information sets intersect, denoted as $C_i \cap C_j$.

## 3.2 Topic-based Prompts

The principal distinction between our Topic-DPR and other prompt-based dense retrieval methods lies in using multiple topic-based prompts to enhance embedding space uniformity and improve retrieval performance. The idea behind creating topic-based prompts is to assign each document a unique prompt that aligns with its semantic and topical diversity. We use semantics, defined by the topic distributions within the simplex space, to initialize the count and values of the topic-based prompts.

We use topic modeling to reveal concealed meanings by extracting topics from a corpus, as explained in Appendix. Topics are defined on a probabilistic simplex (Patterson and Teh, 2013), connecting documents and dictionary words via interpretable probabilistic distributions. We employ hierarchical Latent Dirichlet Allocation (hLDA) (Griffiths et al., 2003), a traditional topic modeling approach, to construct the topic-based prompts. hLDA provides a comprehensive representation of the document collection and captures the hierarchical structure among the topics, which is crucial for seizing the corpus's semantic information diversity.

As shown in Figure 2, hLDA defines hierarchical topic distributions of documents and distributions over words, enabling the generation of topic-based prompts from all hidden topics and corresponding topic words. hLDA creates a hierarchical $K$ topic tree with $h$ levels; each level comprises multiple nodes, each representing a specific topic. This hierarchical structure allows our method to adapt to varying levels of granularity in the topic space, yielding more targeted retrieval results.

Let $\Delta_K$ signify the probabilistic simplex. After uncovering $K$ topics from the corpus, the topic distribution $\theta^{(i)}$ of each document $d_i$ in $\Delta_K$ is defined

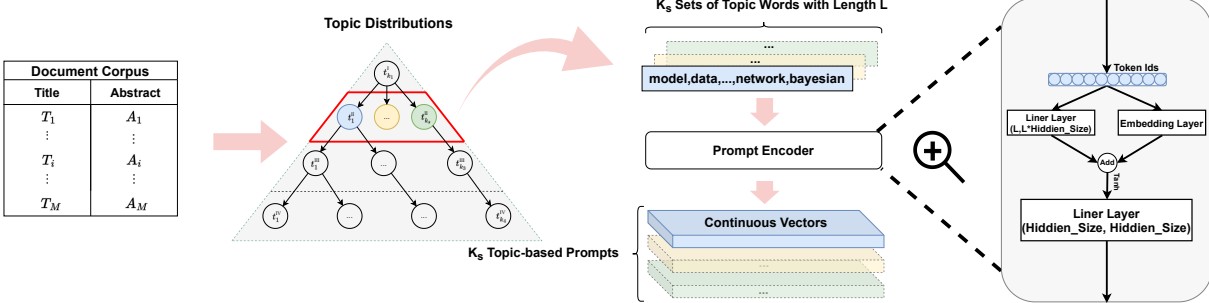

Figure 2: The definition of the topic-based prompts. We model the document corpus using topic modeling to obtain topic distributions. Higher-level $K_s$ topics from the hierarchy are then inputted into the prompt encoder to construct our topic-based prompts. Notably, the parameters of the linear layers within the encoder will be optimized.

as:

$$\theta^{(i)} \in \Delta_K = \left\{ (c_1, ..., c_K) \in R_+^K \mid \sum_{k=1}^{K} c_k = 1 \right\}, \quad (1)$$

where $c_k$ signifies the $k$-th topic's component of $d_i$.

Our topic-based prompts aim to disperse document representations to alleviate semantic space collapse. For these prompts, maintaining significant semantic differences is vital. We only use higher-level $K_s$ topics, a subset of $K$ topics, to form these prompts. These high-level $K_s$ topics are distinctly unique and suitable for defining prompts. Using hLDA, all documents are assigned to one or more topics from the subset in an unsupervised manner, enabling similar documents to share the same topics. This approach enables our method to capture the corpus's inherent topic structure and deliver more accurate and diverse retrieval results.

Each topic $t_k \in \{t_1, ..., t_{K_s}\}$ can be interpreted as a dictionary subset by the top $L$ words with the highest probabilities in $t_k$, defined as $\beta^{(k)} = \{w_1, ..., w_L\}$. We utilize these top $L$ words (topic words) to generate each Topic-based Prompt. We then propose a prompt encoder $E_\Theta$ to embed the discrete word distribution $\beta^{(k)}$, i.e., token ids, into a continuous vector $V_k = E_\Theta(\beta^{(k)})$, assisting PLMs in avoiding local optima during prompt optimization. As shown in Figure 2, $E_\Theta$ primarily comprises a residual network. The embedding layer preserves the topic words' semantic information, and the linear layer represents the trainable parameters $\Theta$. The prompt encoder generates each vector $V_k \in \{V_1, ..., V_{K_s}\}$ as the representation of the topic-based prompt based on topic $t_k$. During retrieval, a document $d_i$ is assigned to a topic-based prompt $P^{(i)} \in \{V_1, ..., V_{K_s}\}$ generated by the topic $t^{(i)}$, where the document has the highest topic component. Documents with similar topic

distributions share the same prompt. For the contrastive learning fine-tuning phase, the PLMs can clearly distinguish simple negative instances with different prompts and focus more on hard negatives with identical prompts.

### 3.3 Topic-DPR with Contrastive Learning

#### 3.3.1 Deep Prompt Tuning

To incorporate our topic-based prompts into the PLMs, we utilize the P-Tuning V2 (Liu et al., 2021b) methodology to initialize a trainable prefix matrix $M$, dimensions $dim \times (num * 2 * dim)$, where $dim$ denotes the hidden size, corresponding to our topic-based prompts in Figure 2, $num$ refers to the transformer layers count, and 2 represents a key vector $K$ and a value vector $V$. These dimensions specifically support the attention mechanism (Vaswani et al., 2017), which operates on key-value pairs and needs to align with the transformer's hidden size and layer structure.

As illustrated in Figure 3 (middle), we encode the title $T_i$ and the assigned prompt $P^{(i)}$ as the query $q_i = Attention[M(P^{(i)}), PLMs(T_i)]$, and the abstract along with its prompt as passage $p_i = Attention[M(P^{(i)}), PLMs(A_i)]$. Each self-attention computation $A_i$ of $Attention[M(P^{(i)}), PLMs(T_i)]$ can be formulated as:

$$\begin{aligned} K_{prompt}, V_{prompt} &\in M(P^{(i)}), \\ Q_{input}, K_{input}, V_{input} &\in PLMs(T_i), \\ K = [K_{prompt}; K_{input}], V &= [V_{prompt}; V_{input}], \\ A_i = Softmax(Q_{input}K^\top)V, \end{aligned} \quad (2)$$

This calculates a weighted sum of input embeddings, i.e., $M(P^{(i)})$ and $PLMs(T_i)$, $M(P^{(i)})$ and $PLMs(A_i)$, based on their contextual relevance. Our approach uses the attention mechanism to

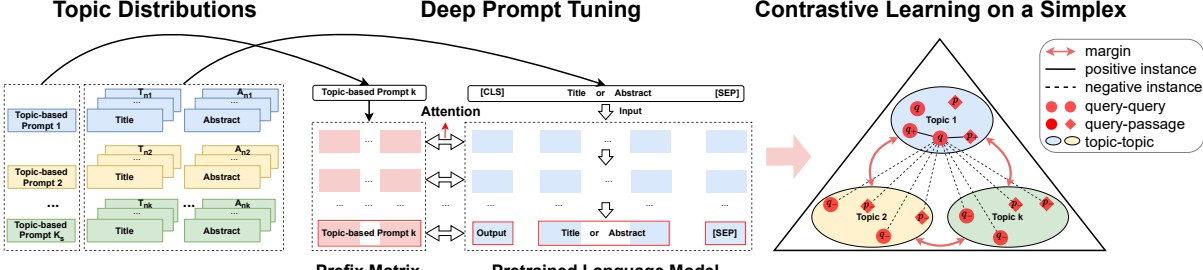

Figure 3: The Topic-DPR framework comprises three main components. First, we associate the document titles or abstracts with topic-based prompts based on their topic distributions (left). Secondly, during the deep prompt tuning phase, the prefix matrix houses the parameters for these prompts, and a pre-trained language model serves as the encoder for titles or abstracts, with the attention mechanism facilitating inter-layer output interactions (middle). Lastly, we introduce three contrastive learning objectives to group relevant queries and passages on a simplex for efficient dense retrieval tasks (right).

amalgamate topic-based prompts with PLM embeddings, enabling focused attention on significant semantic aspects of the input. This facilitates dynamic adjustment of embeddings based on topic-based prompts, leading to more contextually pertinent representations.

Typically, we use the first token $[CLS]$ of output vectors as the query or passage representation. Unlike the Prefix-tuning approach, Topic-DPR simultaneously employs and optimizes multiple prompts, enhancing the model's ability to capture the corpus's diverse semantics, thereby improving retrieval performance.

### 3.3.2 Contrastive Learning

Topic-DPR aims to learn the representations of queries and passages such that the similarity between relevant pairs exceeds that between irrelevant ones. Here, we classify the contrastive loss into three categories: query-query, query-passage, and topic-topic relations.

**Query-Query Relation.** The objective of learning the similarity in the query-query relation is to increase the distance between the negative query $q_i^-$ and the query $q$, while enhancing the similarity between the positive query $q_i^+$ and the query $q_i$. Given a query $q_i$ with $m$ positive queries $\left\{q_{i,z}^+\right\}_{z=1}^m$ and $n$ negative queries $\left\{q_{i,j}^-\right\}_{j=1}^n$, we optimize the loss function as the negative log-likelihood of the query:

$$loss_{\langle q_i, \{q_{i,z}^+\}_{z=1}^m, \{q_{i,j}^-\}_{j=1}^n \rangle} =$$
$$-\frac{1}{m} \sum_{z=1}^m \rho(q_i, q_{i,z}^+) \log \frac{e^{s(q_i, q_{i,z}^+)/\gamma}}{e^{s(q_i, q_{i,z}^+)/\gamma} + \sum_{j=1}^n e^{s(q, q_{i,j}^-)/\gamma}},$$
$$(3)$$

where $\gamma$ is the temperature hyperparameter, and $s(\cdot)$ denotes the cosine similarity function. We define $\rho(\cdot)$ as the correlation coefficient of the positive pairs, which is discussed in Section 3.3.3.

**Query-Passage Relation.** Different from the Eq. 3, the query-passage similarity relation regards the query $q_i$ as the center and pushes the negative passages $\left\{p_{i,j}^-\right\}_{j=1}^n$ farther than the positive passages $\left\{p_{i,z}^+\right\}_{z=1}^m$. Formally, we optimize the loss function as the negative log-likelihood of the positive passage:

$$loss_{\langle q_i, \{p_{i,z}^+\}_{z=1}^m, \{p_{i,j}^-\}_{j=1}^n \rangle} =$$
$$-\frac{1}{m} \sum_{z=1}^m \rho(q_i, p_{i,z}^+) \log \frac{e^{s(q_i, p_{i,z}^+)/\gamma}}{e^{s(q_i, p_{i,z}^+)/\gamma} + \sum_{j=1}^n e^{s(q_i, p_{i,j}^-)/\gamma}},$$
$$(4)$$

Since the objective of the dense passage retrieval task is to find the relevant passages with a query, we consider that the relation of query-passage similarity is critical for the Topic-DPR and the Eq.3 is an auxiliary to Eq.4.

**Topic-Topic Relation.** The motivation for optimizing multiple topic-based prompts lies in the fact that a set of diverse prompts can guide the representations of queries and passages toward the desired topic direction more effectively than a fixed prompt. However, with prompts distributed across $K_s$ topics, the margins between them are still challenging to distinguish using conventional fine-tuning methods. Consequently, we aim to enhance the diversity of these topic-based prompts in the embedding space through contrastive learning to better match their topic distributions. Given a batch of passages $\{A_i\}_{i=1}^N$, we encode them into the PLMs using $K_s$ topic-based prompts $V$ and generate

$K_s \times N$ passages $\left\{ p1^1, ..., p_1^{K_s}, ..., p_N^{K_s} \right\}$, where $p_i^k = Attention[M(V_k), PLMs(A_i)]$. We propose a loss function for each prompt $V_k$ designed to push the other prompts $\{V_z\}_{z \neq k}^{K_s - 1}$ away with the assistance of passages, as formulated below:

$$
loss_{\langle V_k, \{V_z\}_{z \neq k}^{K_s - 1} \rangle} = \frac{1}{(K_s - 1)N^2} \cdot \\
\sum_{z \neq k}^{K_s - 1} \sum_{i,j}^{N} \max \left( \mathcal{M} - s(p_i^k, p_j^k) + s(p_i^k, p_j^z), 0 \right),
$$
(5)

In this function, $\mathcal{M}$ is the margin hyper-parameter, signifying the similarity discrepancy among passages across various topics. It is premised on the belief that unique prompts can steer the same text's representation towards multiple topic spaces. Consequently, Pretrained Language Models (PLMs) can focus on relationships among instances with identical prompts and disregard distractions from unrelated samples with differing prompts. The additional prompts impose constraints on the PLMs, spreading pertinent instances over diverse topic spaces. This approach explains our exclusive use of higher-level topics from hLDA for topic-based prompt definition (Section 3.2).

### 3.3.3 In-batch Positives and Negatives

For dense retrieval, identifying positive and negative instances is vital for performing loss functions Eq.3 and Eq.4. In our approach, we feed a batch of $N$ documents into the PLMs per iteration and sample positives and negatives from this batch. Importantly, we employ multi-category information from these documents to pinpoint relevant queries or passages, aligning with our problem's objective. Queries or passages sharing intersecting multi-category information are considered positive. The correlation coefficient of a positive pair $\langle q_i, q_j^+ \rangle$ can be expressed as:

$$
\rho(q_i, q_j^+) = \frac{|C_i \cap C_i|}{|C_i \cup C_i|},
$$
(6)

This parallels the positive pair in the query-passage relation. By default, all other queries or passages in the batch are deemed irrelevant.

### 3.4 Combined Loss Functions

In this section, we combine the three relations presented above to obtain the combined loss function for fine-tuning over each batch of $\langle \{(q_i, p_i)\}_{i=1}^{N}, \{V_k\}_{k=1}^{K_s} \rangle$:

$$
loss_{total} = \frac{(1 - 2\alpha)}{N} * \sum_{i=1}^{N} loss_{\langle q_i, \{p_{i,z}^+\}_{z=1}^{m}, \{p_{i,j}^-\}_{j=1}^{n} \rangle} \\
+ \frac{\alpha}{N} * \sum_{i=1}^{N} loss_{\langle q_i, \{q_{i,z}^+\}_{z=1}^{m}, \{q_{i,j}^-\}_{j=1}^{n} \rangle} \\
+ \frac{\alpha}{K_s} * \sum_{k=1}^{K_s} loss_{\langle V_k, \{V_z\}_{z \neq k}^{K-1} \rangle},
$$
(7)

where $\alpha$ is a hyper-parameter to weight losses.

## 4 Experiments

### 4.1 Experimental Settings

**Datasets** We evaluate Topic-DPR's retrieval performance through experiments on two scientific document datasets: the arXiv-Article (Clement et al., 2019) and USPTO-Patent datasets (Li et al., 2022a). For dense passage retrieval, we extract titles, abstracts, and multi-category information from these semi-structured datasets, using titles as queries and relevant abstracts as optimal answers. Appendix details these datasets' statistics.

**Evaluation Metrics** Considering the realistic literature retrieval process and dataset passage count, we use Accuracy (Acc@1, 10), Mean Reciprocal Rank (MRR@100), and Mean Average Precision (MAP@10, 50) for performance assessment. Furthermore, we apply a representation analysis tool to gauge the alignment and uniformity of the PLMs embedding space (Wang and Isola, 2020).

**Baselines** For a baseline comparison, we employ the standard sparse retrieval method, BM25 (Robertson et al., 2009). The efficacy of our proposed techniques is assessed against DPR and DPTDR, adapted to our datasets. DPR, an advanced dual-encoder method, incorporates contrastive learning for dense passage retrieval, while DPTDR, a contemporary leading method using deep prompt tuning, employs a continuous prompt to boost PLMs' retrieval efficiency. Due to the absence of specific positive examples for each query, we apply the positive sampling approach across all techniques to guarantee fair comparisons.

**Implementation Details** We initialize our Topic-DPR parameters using two uncased PLMs, BERT-base, and BERT-large, obtained from Huggingface. All experiments are executed on Sentence-Transformer with an NVIDIA Tesla A100 GPU. Appendix details all the hyper-parameters.

| Methods | PLM Frozen | Acc@1 | Acc@10 | MRR@100 | MAP@10 | MAP@50 |
|---|---|---|---|---|---|---|
| BM25 | - | 80.20/80.72 | 96.34/96.70 | 85.98/86.41 | 31.48/35.38 | 16.18/20.26 |
| Experiments on BERT-base | | | | | | |
| DPR | ✘ | 90.98/88.96 | 98.26/97.87 | 93.59/92.27 | 57.86/54.70 | 49.53/44.53 |
| DPTDR | ✘ | 90.75/88.62 | 97.68/97.73 | 93.43/92.04 | 58.57/55.66 | 50.05/45.71 |
| **Topic-DPR** | ✘ | **91.40/90.29** | **98.43/98.32** | **94.21/93.26** | **61.57/58.08** | **52.69/48.63** |
| DPTDR♠ | ✔ | 88.00/86.11 | 98.02/97.06 | 91.82/90.29 | 50.83/50.04 | 38.45/37.92 |
| **Topic-DPR♠** | ✔ | **89.19/87.41** | **98.13/97.63** | **93.01/91.16** | **53.19/52.68** | **41.39/40.51** |
| Experiments on BERT-large | | | | | | |
| DPR | ✘ | 91.44/89.06 | 98.68/97.88 | 94.37/92.42 | 58.92/56.28 | 51.44/46.07 |
| DPTDR | ✘ | 91.12/87.93 | 98.67/97.61 | 94.01/91.61 | 59.62/57.50 | 52.16/47.33 |
| **Topic-DPR** | ✘ | **91.98/91.04** | **98.72/98.50** | **94.50/93.31** | **63.01/61.18** | **54.71/50.29** |
| DPTDR♠ | ✔ | 90.25/87.32 | 98.34/97.44 | 93.35/91.22 | 54.09/52.89 | 41.79/41.18 |
| **Topic-DPR♠** | ✔ | **91.03/90.55** | **98.40/98.40** | **93.76/92.84** | **56.83/54.25** | **45.96/43.32** |

Table 1: Experimental results on the arXiv-Article and USPTO-Patent datasets in dense passages retrieval tasks (results presented as arXiv-Article/USPTO-Patent). And ♠ indicates that only the parameters of prompts can be tuned and the the parameters of PLM are frozen.

## 4.2 Experimental Results

Table 1 presents our experiments' outcomes using BERT-base and BERT-large models on the arXiv-Article and USPTO-Patent datasets. In comparison to sparse methods, dense retrieval techniques show significant performance improvement, emphasizing dense retrieval's importance. When contrasting DPTDR with DPR, DPTDR exhibits superior performance in the MAP@10 and MAP@50 metrics due to its continuous prompt enhancement. Our topic-based prompts in Topic-DPR boost the Acc and MRR metrics. Furthermore, Topic-DPR outperforms baseline methods across all metrics. Specifically, our Topic-DPR$base$ exceeds DPTDR$base$ by 3.00/2.42 and 2.64/2.98 points in MAP@10 and MAP@50, which are vital for large multi-category passage retrieval. Additionally, in the deep prompt tuning setting, our Topic-DPR♠, despite slight performance degradation, still maintains comparative performance with only 0.1%-0.4% of the parameters tuned. The consistent enhancements across diverse settings, models, and metrics manifest the robustness and efficiency of our Topic-DPR method. This research establishes Topic-DPR as an effective deep prompt learning-based dense retrieval method, setting a new state-of-the-art for the datasets. Ablation experiments are conducted in Appendix.

## 5 Analysis on Topic-DPR

| Methods | Acc@10 | MRR@100 | MAP@50 |
|---|---|---|---|
| DPR | 98.26 | 93.59 | 49.53 |
| DPR with random words | 96.24 | 89.19 | 48.64 |
| DPR with topic words | 98.30 | 93.73 | 50.19 |
| **Topic-DPR** | **98.43** | **94.21** | **52.69** |

Table 2: Quality analysis of the topic words on the test set of the arXiv-Article dataset. DPR with topic words indicates that each example in the training data has the corresponding topic words added as prompts. DPR with random words indicates that each example in the training data has random topic words added as prompts.

## 5.1 Quality of Topic Words

To assess whether the topic words extracted from hLDA are helpful for dense passage retrieval tasks, we conducted an experiment that directly used the topic words as prompts to train DPR with BERT-base. Each query was transformed into "[TOPIC WORDS...] + [QUERY]" and each passage was transformed into "[TOPIC WORDS...] + [PASSAGE]". As shown in Table 2, when noise is introduced, the model experiences more interference, resulting in decreased performance. This indicates that simply adding random words to the queries and passages is detrimental to the model's ability to discern relevant information. However, with the help of the topical information extracted from hLDA, the model performs better than the original DPR. This suggests that introducing high-quality topic words is beneficial for domain disentanglement, as it allows the model to better differentiate between various subject areas and focus on the relevant con-

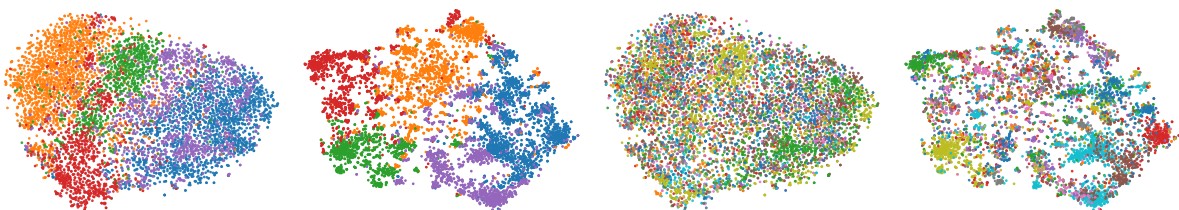

(a) BERT with 5 topics.    (b) Topic-DPR with 5 topics.    (c) BERT with 412 categories.    (d) Topic-DPR with 412 categories.

Figure 4: T-SNE visualization (Van der Maaten and Hinton, 2008) of the representations from the vanilla BERT-base and the Topic-DPR-base. Each point represents one passage under a specific label, and each color represents one class. The two types of labels we use are the five topics generated by the hLDA and the 412 categories in the USPTO-Patent dataset.

text. By incorporating topic words as prompts, the model is able to generate more diverse and accurate semantic representations, ultimately improving its ability to retrieve relevant passages.

## 5.2 Representations with Topic-based Prompts

We visualize the representations with different types of labels and analyze the influence of topic-based prompts intuitively. As shown in Figure 4(a), the passages distributed in $K_a$ topics overlap in the vanilla BERT. As depicted in Figure 4(b), they are arranged in a pentagonal shape, resembling the probabilistic simplex of the topic model $\Delta_K$. Here, each topic is independent and discrete, with clearly distinguishable margins between them. This outcome is primarily due to the topic-based prompts, which are initialized by the topic distributions and direct the representations to exhibit significant topical semantics. During the dense retrieval phase, the passages in sub-topic spaces are more finely differentiated using loss functions Eq.3 and Eq.4. As illustrated in Figure 4(d), passages belonging to the same category cluster together, enabling queries to identify relevant passages with greater accuracy. These observations suggest that our proposed topic-based prompts can encourage PLMs to generate more diverse semantic representations for dense retrieval.

## 5.3 Alignment and Uniformity

We experiment with 5,000 pairs of queries and passages from the USPTO-Patent dataset's development set to analyze the quality of representations in terms of three metrics, including alignment, uniformity, and cosine distance. Alignment and uniformity are two properties to measure the quality of representations (Wang and Isola, 2020). Specifi-

| Methods | Align(q,p$^+$) | Uniform(q)/(p) | Sim(q,p$^+$) /(q,p$^-$) |
|---|---|---|---|
| BERT | 0.73 | -0.72/ -0.98 | 63.71 /58.45 |
| DPR | 0.47 | -2.60/ -2.22 | 78.90 / 29.46 |
| DPTDR | 0.42 | -2.50/ -2.13 | 79.89 / 34.57 |
| **Topic-DPR** | **0.38** | **-3.08/ -3.02** | **80.67/ 14.81** |

Table 3: Quality analysis of the representations generated by different methods in BERT-base. The quality is better when all the above numbers are lower except Sim(q,p$^+$).

cally, the alignment measures the expected distance between the representations of the relevant pairs $(x, x^+)$:

$$\text{Align}(x, x^+) \triangleq \underset{(x_i, x_i^+) \backsim (x, x^+)}{\mathbb{E}} \left\| f(x_i) - f\left(x_i^+\right) \right\|^2,$$
(8)

where the $f(x)$ is the L2 normalization function. And the uniformity measures the degree of uniformity of whole representations $p_{\text{data}}$ :

$$\text{Uniform}(p_{\text{data}}) \triangleq \log \underset{\substack{x, y \underset{i.i.d.}{\mathbb{E}} p_{\text{data}}}}{\mathbb{E}} e^{-2\|f(x)-f(y)\|^2},$$
(9)

As demonstrated in Table 3, the results of all dense retrieval methods surpass those of the vanilla BERT. Comparing DPR with DPTDR, the latter exhibits better alignment performance yet poorer uniformity, with representations from DPR displaying significant differences in average similarity between relevant and irrelevant pairs. This phenomenon highlights a shortcoming of deep prompt tuning in dense retrieval, where the influence of a single prompt can lead to anisotropic issues in the embedding space. Furthermore, this observation can explain why DPTDR underperforms DPR in certain metrics, as discussed in Section 4.2. Regarding the results of our Topic-DPR, the alignment of representations decreases to 0.38, marginally

better than DPTDR. Importantly, our method substantially enhances uniformity, achieving the best scores and indicating a greater separation between relevant and irrelevant passages. Thus, our Topic-DPR effectively mitigates the anisotropic issue of the embedding space.

## 6 Conclusion

In this paper, we examine the limitations of using a single task-specific prompt for dense passage retrieval. To address the anisotropic issue in the embedding space, we introduce multiple novel topic-based prompts derived from the corpus's semantic diversity to improve the uniformity of the space. This approach proves highly effective in identifying and ranking relevant passages during retrieval. We posit that the strategy of generating data-specific continuous prompts may have broader applications in NLP, as these prompts encourage PLMs to represent more diverse semantics.

## Limitations

Our method achieves promising performance to enhance the semantic diversity of representations for dense passage retrieval, but we believe that there are two limitations to be explored for future works: (1) The topic modeling based on the neural network may be joint trained with the dense passage retrieval task, where the topics extracted for our topic-based prompts can be determined to comply with the objective of retrieval automatically. (2) The possible metadata of documents, like authors, sentiments and conclusions, can be considered to assist the retrieval of documents further.

## Acknowledgements

This work was supported by National Natural Science Foundation of China (No. 62006083), Natural Science Foundation of Guangdong (2023A1515012073) and National Key Research and Development Program of China (2020YFA0712500).

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
