# OpenReview forum: "Topic-DPR: Topic-based Prompts for Dense Passage Retrieval"
_EMNLP/2023/Conference — EMNLP 2023 Findings_

### Official Review · Reviewer_q4iy · 2023-07-24

**Soundness:** 3

**Excitement:**

2: Mediocre: This paper makes marginal contributions (vs non-contemporaneous work), so I would rather not see it in the conference.

**Missing References:**

[1] Izacard, Gautier, Mathilde Caron, Lucas Hosseini, Sebastian Riedel, Piotr Bojanowski, Armand Joulin and Edouard Grave. “Unsupervised Dense Information Retrieval with Contrastive Learning.” Trans. Mach. Learn. Res. 2022 (2021): n. pag.
[2] Ni, Jianmo, Chen Qu, Jing Lu, Zhuyun Dai, Gustavo Hernandez Abrego, Ji Ma, Vincent Zhao, Yi Luan, Keith B. Hall, Ming-Wei Chang and Yinfei Yang. “Large Dual Encoders Are Generalizable Retrievers.” Conference on Empirical Methods in Natural Language Processing (2021).
[3] Su, Hongjin, Weijia Shi, Jungo Kasai, Yizhong Wang, Yushi Hu, Mari Ostendorf, Wen-tau Yih, Noah A. Smith, Luke Zettlemoyer and Tao Yu. “One Embedder, Any Task: Instruction-Finetuned Text Embeddings.” Annual Meeting of the Association for Computational Linguistics (2022).

**Paper Topic And Main Contributions:**

This paper proposes topic-aware prompt learning for passage retrieval.
Authors recognize that previous works focus on only optimizing a single vector for passage retrieval, which can lead to a semantic space collapse. Hence, in this paper, they propose to use topic-based prompts for passage retrieval. Specifically, they employ a topic model to mine topics in passages. Then these topics will improve the passage retrieval by contrastive learning.


**Questions For The Authors:**

Question A:
How can different topic mining models affect passage retrieval? (Robustness to topic modeling.)

**Reasons To Accept:**

1. This paper is well-written and easy to follow.
2. Authors propose to aggregate topics into prompts for prompt learning to improve passage retrieval. The motivation of this paper makes sense and is well-described in the paper.
3. The results shown in Table 1 is promising. This method could achieve non-trivial improvement compared to baselines.


**Reasons To Reject:**

1. More baselines are needed to validate the effectiveness of the proposed methods. Only two dense retrieval baselines are included in the experiments and other methods introduced in related work are not selected as baselines in experiments.
2. Some related works about dense retrieval for text are not mentioned in this paper, such as Contriever [1], GTR [2] and Instructor [3], etc.
[1] Izacard, Gautier, Mathilde Caron, Lucas Hosseini, Sebastian Riedel, Piotr Bojanowski, Armand Joulin and Edouard Grave. “Unsupervised Dense Information Retrieval with Contrastive Learning.” Trans. Mach. Learn. Res. 2022 (2021): n. pag.
[2] Ni, Jianmo, Chen Qu, Jing Lu, Zhuyun Dai, Gustavo Hernandez Abrego, Ji Ma, Vincent Zhao, Yi Luan, Keith B. Hall, Ming-Wei Chang and Yinfei Yang. “Large Dual Encoders Are Generalizable Retrievers.” Conference on Empirical Methods in Natural Language Processing (2021).
[3] Su, Hongjin, Weijia Shi, Jungo Kasai, Yizhong Wang, Yushi Hu, Mari Ostendorf, Wen-tau Yih, Noah A. Smith, Luke Zettlemoyer and Tao Yu. “One Embedder, Any Task: Instruction-Finetuned Text Embeddings.” Annual Meeting of the Association for Computational Linguistics (2022).
3. BERT in Figure 4 is not a good baseline to compare. BERT is not designed to directly represent passages and then has low-quality passage representations. Authors should at least compare the representations from DPR and DPTDR.

**Reproducibility:**

4: Could mostly reproduce the results, but there may be some variation because of sample variance or minor variations in their interpretation of the protocol or method.

**Reviewer Confidence:**

3: Pretty sure, but there's a chance I missed something. Although I have a good feel for this area in general, I did not carefully check the paper's details, e.g., the math, experimental design, or novelty.

---

> ### Author Rebuttal · Authors · 2023-08-28
>
> Thanks for your comments.
> > For Question A:
> How can different topic mining models affect passage retrieval? (Robustness to topic modeling.)
>
> Answer Q-A:
> 1. Currently, there are many topic mining models available, including those based on Bayesian theory and neural networks. I have also used LDA and GPT models to extract topics in our framework, and it is undeniable that they each have their own advantages. For neural network-based topic mining models, they are all trained on large-scale domain-specific data sets, making cross-domain transfer difficult. As for the Bayesian-based model, its advantage lies in probability simplex which gives our method great interpretability. Moreover, this model is highly practical and requires only low time and training costs to complete cross-domain training transfer; therefore we choose the Bayesian-based model.
>
> 2. Compared with the most popular LDA model, this model requires setting many hyperparameters which may affect our framework's performance more or less. Therefore we choose hLDA (hierarchical LDA) model which inherits the advantages of LDA while having fewer hyperparameters to automatically obtain multiple hierarchical structure topics.
>
> 3. From our motivation perspective, we use topic modeling to establish different subspaces for further comparative learning training to cluster data sets more accurately. The hLDA model can automatically extract K high-level topics to help define prompts that meet requirements.
>
> 4. In terms of practicality and ease of operation, we use the hLDA model and demonstrate its robustness effects in datasets from different domains.
>
> > For Rejection 1:
> More baselines are needed to validate the effectiveness of the proposed methods. Only two dense retrieval baselines are included in the experiments and other methods introduced in related work are not selected as baselines in experiments.
>
> Answer R1:
> Thank you for your feedback. We will compare more baselines to improve the traditional prefix-tuning mechanism and enhance retrieval performance. Therefore, we chose these two reproducible and directly comparable baselines. We conducted a fair comparison of the baseline model to demonstrate the advantages of our proposed topical prompt-tuning. As for other baseline strategies for sampling positive and negative examples, they can be easily integrated into our framework.
>
> > For Rejection 2:
> Some related works about dense retrieval for text are not mentioned in this paper, such as Contriever [1], GTR [2] and Instructor [3], etc.
>
> Answer R2:
> Thank you, we will cite these articles in the next version.
>
> > For Rejection 3:
> BERT in Figure 4 is not a good baseline to compare. BERT is not designed to directly represent passages and then has low-quality passage representations. Authors should at least compare the representations from DPR and DPTDR.
>
> Answer R3:
> We will add the graphs of DPR and DPTDR in the next version. This experiment aims to demonstrate whether the vector distribution of our model conforms to the probability simplex, which is more intuitive compared with BERT before and after training. Due to numerous categories and limited space in the paper, it is difficult to compare DPR and DRTDR directly, but their results can be observed from the experimental table.

---

### Official Review · Reviewer_ZAsx · 2023-07-31

**Soundness:** 4

**Excitement:**

4: Strong: This paper deepens the understanding of some phenomenon or lowers the barriers to an existing research direction.

**Paper Topic And Main Contributions:**

This paper explores the use of multiple prompts for more nuanced sub-topics or specialized areas. In particular, the authors adopt topic modeling techniques to generate more fine-grind topics and combine it with deep retrievers which are aware of contextualized representations. The main contributions of this paper include the introduction of topic modeling for training better retrievers by categorizing document topics and alleviating the anisotropic similarity representations generated by deep retrieval models.

**Reasons To Accept:**

This paper proposed a novel method that not only boosts the retrieval effectiveness, but also justified the representation improvement by alignment and uniformity experiments quantitively. The anisotropic representation is a long-standing issue raised by training deep retrievers. This makes the proposed work more sound for general interests of neural retrievers.

Another good part of this paper is successful introduction of a traditional topic modeling method for the IR domain. Using contextualized deep representations with topic modeling looks promising.

**Reasons To Reject:**

1) Some important technical details are missing. For example, how negative query pairs and query-passage pairs are selected. If these are provided in the datasets, please specify.

2) Very limited baseline systems (DPR and DPTDR) are compared, and the popularity of the datasets arXiv-Article and USPTO-Patent prevents the authors from comparing more systems. I would suggest evaluating additional (more popular) systems tuning by topics (e.g., TAS-B [1]) and show more convincing results. (Since your topic clustering is unsupervised, why not apply it to more popular datasets?)

3) Missing further ablations to study why simple DPR with topic words are worse, are DPR w/ topic words using the same loss function? Is the gap due to the additional parameters from the trainable matrix M? Showing the two are compared fairly and the difference is made from representation uniformity would be desired, and it will align to the center conclusion of this paper.

4) Notations may be too verbose to follow, e.g., what is the difference between P(i) and Vk? Why it defines ∆K (in simplex space), but later I do not see it is used. Is beta^(k) in simplex space? In short, the paper needs some refinements on the formulas to make it easier to follow.

[1] Efficiently Teaching an Effective Dense Retriever with Balanced Topic Aware Sampling -- Sebastian Hofstätter et al.

**Reproducibility:**

4: Could mostly reproduce the results, but there may be some variation because of sample variance or minor variations in their interpretation of the protocol or method.

**Reviewer Confidence:**

3: Pretty sure, but there's a chance I missed something. Although I have a good feel for this area in general, I did not carefully check the paper's details, e.g., the math, experimental design, or novelty.

---

> ### Author Rebuttal · Authors · 2023-08-28
>
> Thank for your hard work and valuable comments.
>
> > For Rejection 1:
> Some important technical details are missing. For example, how negative query pairs and query-passage pairs are selected. If these are provided in the datasets, please specify.
>
> Answer R1:
> Queries or passages within the same batch that share intersecting multi-category information from arxiv or patent datasets are considered positive. For instance, here are three document samples: A (categories 1, 2, and 3), B (categories 2 and 4), and C (category 5). In this case, we can consider pairs such as (A,A) and (A,B) as positive pairs for A while considering (A,C) as a negative pair. For a fair comparison, we have used the same strategy in all the baselines for comparison.
>
>
> > For Rejection 2:
> Very limited baseline systems (DPR and DPTDR) are compared, and the popularity of the datasets arXiv-Article and USPTO-Patent prevents the authors from comparing more systems. I would suggest evaluating additional (more popular) systems tuning by topics (e.g., TAS-B [1]) and show more convincing results. (Since your topic clustering is unsupervised, why not apply it to more popular datasets?)
>
> Answer R2:
> Thank you, our method of selecting positive samples is similar to TAS-B and we will increase the reference and comparison. From the perspective of the dataset, we used multi-class information as one of the selection criteria for positive samples, which exists in both semi-structured datasets arXiv-Article and USPTO-Patent. From a baseline perspective, since we are innovating from the perspective of prefix fine-tuning, we have chosen DPR and DPTDR as two baselines that are easy to reproduce and directly comparable. For other baseline strategies for sampling positive and negative examples, they can be directly integrated with our framework.
>
>
> > For Rejection 3:
> Missing further ablations to study why simple DPR with topic words are worse, are DPR w/ topic words using the same loss function? Is the gap due to the additional parameters from the trainable matrix M? Showing the two are compared fairly and the difference is made from representation uniformity would be desired, and it will align to the center conclusion of this paper.
>
> Answer R3:
> According to Table 2, the DPR with topic words outperforms the simple DPR, and they utilize the same loss function. However, there still exists a gap when compared to Topic-DPR. In response to your question, "why is the simple DPR with topic words inferior to Topic-DPR? Is the gap attributed to the additional parameters from the trainable matrix M?", we offer the following analysis:
>
> 1. The trainable matrix M proves effective for training but without topic words. DPTDR serves as an example of this. It enhances the DPR by introducing additional parameters from M and uses a fixed set of vectors as input, which are randomly determined.
>
> 2. Without designing a specific loss function, the trainable matrix M becomes ineffective or even detrimental when integrating multiple sets of topic words. Firstly, since our trainable matrix M is randomly initialized and hasn't undergone Masked Language Model training, it diverges from DPR w/ topic words. The latter can recognize topic words due to its foundational weights. In contrast, the trainable matrix  M lacks semantic comprehension for multiple sets of topic word vectors. In other words, inputting multiple sets of topic word vectors is equivalent to inputting multiple sets of random vectors. For DPR with trainable matrix M and topic words (i.e. DPTDR with topic words), the trainable matrix M faces greater challenges during fitting, resulting in performance inferior to that of a fixed set of random vectors (i.e. DPTDR). Directly using multiple sets of topic word vectors as inputs for the trainable matrix M doesn't make sense. Similar conclusions can be drawn from the ablation study in the appendix, specifically "Figure 5, Topic-DPR w/o TTR", where Topic-DPR without the Topic-Topic Relations training objection is worse than DPTDR. Thus, this performance gap stems from the contrastive learning training loss designed for topic-to-topic relations, represented by equation 5, "topic-topic relations". This loss aims to train a topic space that resembles a probability simplex.
>
>
> > For Rejection 4:
> Notations may be too verbose to follow, e.g., what is the difference between P(i) and Vk? Why it defines ∆K (in simplex space), but later I do not see it is used. Is beta^(k) in simplex space? In short, the paper needs some refinements on the formulas to make it easier to follow.
>
> Answer R4:
> 1. We will redefine it easier to follow. The matrix P(i) is obtained by multiplying the matrix Vk with the trainable matrix M. The \theta(i) in ∆K help us define the topical prompts beta^(k) in such simplex space, and then built the simplex space in the bert embedding space with the topic-topic relations training objective. Beta^(k) can be considered as the vector representation of each topic in ∆K. And also beta^(k) bring the initial semantic via word embedding layer, represented by the right of figure 2. We have already used it.
>
> 2. From Figure 4, it can be seen that the distribution of vectors conforms to the high-level topical distributions in simplex space ∆K as defined by us, which is closely related to our proposed training method for topic prompts.
>
> 3. About the theory of probability simplex, you can refer to this article "Sam Patterson and Yee Whye Teh. 2013. Stochastic gradient Riemannian Langevin dynamics on the probability simplex. Advances in neural information processing systems, 26."

---

### Official Review · Reviewer_P8b2 · 2023-08-05

**Soundness:** 3

**Excitement:**

3: Ambivalent: It has merits (e.g., it reports state-of-the-art results, the idea is nice), but there are key weaknesses (e.g., it describes incremental work), and it can significantly benefit from another round of revision. However, I won't object to accepting it if my co-reviewers champion it.

**Paper Topic And Main Contributions:**

This paper proposes a continue prompt tuning approach with multiple topic-based prompts in dense passage retrieval to overcome the limitation of single prompt tuning, that help PLM model to capture diverse semantic information of dataset. Hierarchical LDA is used to construct the topic-based prompts, specifically for each topic top L words with highest probabilities have been used to generated the prompt respectively. Authors adopt existing prefix-tuning technique and concatenate topic-based prompts with PLM embeddings in attention computation. Furthermore, to train the model to learn query and passages embeddings, they compute contrastive loss in three aspects: query-query, query-passage, and topic-topic.

**Questions For The Authors:**

1. It would be better to understand the importance of three contrastive loss if there are some ablation study on it.
2. Why is performance difference between model and baselines lower in Acc@K than MAP@K?


**Reasons To Accept:**

1. This paper introduces a novel data-driven way to compute multiple topic-based continuous prompts, that help model to get rid of semantic space collapse caused by using a single prompt. The multiple prompts can capture the distinct diverse semantics information through topic-distribution in the dataset. As a result, the passage and query representation can be well distributed in the latent space, and so query relevant passages can be easier to identify.

2.  The problem is well motivated.

**Reasons To Reject:**

1. In the experimental results, Topic-DPR achieve marginal improvement over the performance of simple DPR model, specifically Acc@K. That implies simple way of first extracting topic words from the dataset, then using them in prompts might not be sufficient enough.

2. The novelty of this paper is limited to incorporating an existing topic model with prefix tuning approach.

**Reproducibility:**

3: Could reproduce the results with some difficulty. The settings of parameters are underspecified or subjectively determined; the training/evaluation data are not widely available.

**Reviewer Confidence:**

3: Pretty sure, but there's a chance I missed something. Although I have a good feel for this area in general, I did not carefully check the paper's details, e.g., the math, experimental design, or novelty.

---

> ### Author Rebuttal · Authors · 2023-08-28
>
> > For Question 1:
>  It would be better to understand the importance of three contrastive loss if there are some ablation study on it.
>
> Answer Q1:
> We conducted an ablation study on three types of contrastive losses. Due to the length limitation of the paper, we have included them in the appendix.
>
>
> > For Question 2:
> Why is performance difference between model and baselines lower in Acc@K than MAP@K?
>
> Answer Q2:
> 1. Firstly, for the baseline models, the Acc@K score already lies between 90%-98%, a notably high accuracy level. With such elevated accuracy, achieving any further significant improvements becomes extremely challenging, and a smaller performance gap is anticipated.
>
> 2. In retrieval scenarios, Acc@K primarily focuses on the presence of the correct answer within the top K results, without giving due consideration to the specific ordering of these answers. This renders Acc@K a relatively simplistic evaluation metric, unable to holistically represent the overall ranking quality of retrieval results. Consequently, when juxtaposed against baselines, our model finds it challenging to demonstrate a marked difference in Acc@K.
>
> 3. In contrast to Acc@K, MAP@K is considerably more stringent. It not only emphasizes the presence of the correct answers but also places significant importance on their order. This ensures that MAP@K provides a more granular assessment of the retrieval system's performance. The substantial improvement we observed in MAP@K underscores the effectiveness and superiority of our proposed method.
>
>
> > For Rejection 2:
> The novelty of this paper is limited to incorporating an existing topic model with prefix tuning approach.
>
> Answer R2:
> 1. Our approach is effective and non-trivial not only because of the combination of topic modeling and prefix fine-tuning, but also because we designed a contrastive learning training objective for inter-topic relations based on extracted topic prompts and retrieval scenarios, as shown in Formula 5 Topic-Topic Relation. There are two additional experiments that prove this point: Firstly, Section 5.1 demonstrates the effectiveness of topic cues but with some gaps compared to our method due to Topic-Topic Relation; Secondly, the ablation experiment in the appendix also proves the effectiveness of this training objective.
>
> 2. According to Table 2, the DPR with topic words outperforms the simple DPR, and they utilize the same loss function. However, there still exists a gap when compared to Topic-DPR. In response to Reviewer ZAsx's question, "why is the simple DPR with topic words inferior to Topic-DPR? Is the gap attributed to the additional parameters from the trainable matrix M?", we offer the following analysis:
>
>     2.1 Firstly, the trainable matrix M proves effective for training but without topic words. DPTDR serves as an example of this. It enhances the DPR by introducing additional parameters from M and uses a fixed set of vectors as input, which are randomly determined.
>
>     2.2 Secondly, without designing a specific loss function, the trainable matrix M becomes ineffective or even detrimental when integrating multiple sets of topic words. Firstly, since our trainable matrix M is randomly initialized and hasn't undergone Masked Language Model training, it diverges from DPR w/ topic words. The latter can recognize topic words due to its foundational weights. In contrast, the trainable matrix  M lacks semantic comprehension for multiple sets of topic word vectors. In other words, inputting multiple sets of topic word vectors is equivalent to inputting multiple sets of random vectors. For DPR with trainable matrix M and topic words (i.e. DPTDR with topic words), the trainable matrix M faces greater challenges during fitting, resulting in performance inferior to that of a fixed set of random vectors (i.e. DPTDR). Directly using multiple sets of topic word vectors as inputs for the trainable matrix M doesn't make sense. Similar conclusions can be drawn from the ablation study in the appendix, specifically "Figure 5, Topic-DPR w/o TTR", where Topic-DPR without the Topic-Topic Relations training objection is worse than DPTDR. Thus, this performance gap stems from the contrastive learning training loss designed for topic-to-topic relations, represented by equation 5, "topic-topic relations". This loss aims to train a topic space that resembles a probability simplex.
>
> 3. Topic modeling is an effective and transferable technique, while prefix fine-tuning is currently popular. Therefore, I believe that our work demonstrating the effectiveness of combining these two methods is also necessary.

---

### Meta-Review · Area_Chair_rVcZ · 2023-09-19

**Recommendation:** 4

**Metareview:**

This paper presents an interesting idea of topic-based prompt learning for sense passage retrieval. The idea is well motivated. The proposed model incorporates well the idea. The experiments demonstrate the strength of the proposed approach against a single prompt.

The reviewers have found the idea new and interesting. The authors have answered most of the questions raised in the reviews in their rebuttal. It is expected that some of the replies will be added into the final version.

Overall, the paper proposes a new and interesting idea for topic-dependent prompt. The paper may inspire other researchers in the future.

---

### Decision · Program_Chairs · 2023-10-07

**Decision:**

Accept-Findings

**Comment:**

This paper presents an interesting idea of topic-based prompt learning for sense passage retrieval. The idea is well motivated. The proposed model incorporates well the idea. The experiments demonstrate the strength of the proposed approach against a single prompt.

The reviewers have found the idea new and interesting. The authors have answered most of the questions raised in the reviews in their rebuttal. It is expected that some of the replies will be added into the final version.

Overall, the paper proposes a new and interesting idea for topic-dependent prompt. The paper may inspire other researchers in the future.